
# A Comparison of Two Nonlinear Data Assimilation Methods

Vivian A. Montiforte[1,2], Hans E. Ngodock[1], and Innocent Souopgui[1,3]

[1]U.S. Naval Research Laboratory, 1009 Balch Boulevard, Stennis Space Center, MS 39529
[2]American Society for Engineering Education, 1818 N Street N.W. Suite 600, Washington, D.C. 20036
[3]Redline Performance Solutions, LLC., 9841 Washingtonian Blvd, Suite 200, Gaithersburg, MD 20878

**Correspondence:** Vivian A. Montiforte (vivian.montiforte.ctr@nrlssc.navy.mil)

**Abstract.** Advanced numerical data assimilation (DA) methods, such as the four-dimensional variational (4DVAR) method, are elaborate and computationally expensive. Simpler methods exist that take time-variability into account, providing the potential of accurate results with a reduced computational cost. Recently, two of these DA methods were proposed for a nonlinear ocean model. The first method is Diffusive Back and Forth Nudging (D-BFN) which has previously been implemented in several

complex models, most specifically, an ocean model. The second is the Concave-Convex Nonlinearity (CCN) method provided by Larios and Pei that has a straightforward implementation and promising results. D-BFN is less costly than a traditional variational DA system but it requires integrating the model forward and backward in time over a number of iterations, whereas CCN only requires integration of the forward model once. This paper will investigate if Larios and Pei's CCN algorithm can provide competitive results with the already tested D-BFN within simple chaotic models. Results show that observation density

and/or frequency, as well as the length of the assimilation window, significantly impact the results for CCN, whereas D-BFN is fairly adaptive to sparser observations, predominately in time.

## 1 Introduction

There are generally two classes of data assimilation (DA) methods: filters and smoothers. The filters, also referred to as

sequential methods, compute a DA analysis at a selected time (called the analysis time), given a model background state (or forecast state) and data collected during a period of time up to the analysis time (observation window). Commonly used filters include the three-dimensional variational (3DVAR) (Barker et al., 2004; Daley and Barker, 2001; Lorenc, 1981; Lorenc et al., 2000), the Kalman Filter (Kalman, 1960), and the ensemble Kalman Filter (EnKF) (Evensen, 1994), along with its many variants. Filters assume that all the data within the observation window are collected and valid at the analysis time. Although

this assumption may be warranted for slowly evolving processes and short observation windows, it has the undesirable effect of assimilating observations at the wrong time and suppressing the time variability in the observations (if multiple observations are collected at the same location within the observation window, only one of them will be assimilated).





Smoothers on the other hand assimilate all observations collected within the observation window at their respective time and provide a correction to the entire model trajectory over the assimilation window. Note that there can be a difference between the assimilation window and the observation window. The former refers to the time window over which a correction to the model is computed, while the latter refers to the time window over which observations are collected/considered for assimilation. Smoothers do not have the problem of neglecting observations collected at the same location and different times, which means they do account for the time variability in the observations. However, they are computationally much more expensive than the filters. There are a few known smoother methods such as the four-dimensional variational (4DVAR) (Fairbairn et al., 2013; Le Dimet and Talagrand, 1986), the Kalman Smoother (KS) (Bennett and Budgell, 1989), and the Ensemble Kalman Smoother (EnKS) (Evensen and Van Leeuwen, 2000). Of these three, 4DVAR is the one that is most used in geosciences problems. It does, however, require the development of a tangent linear (TLM) and adjoint of the dynamical model being used. This development of the TLM and the adjoint model is both cumbersome and tedious.

Auroux and Blum proposed a smoother method called Back and Forth Nudging (BFN) (Auroux and Blum, 2005, 2008; Auroux and Nodet, 2011). It consists of nudging the model to the observations in both the forward and backward (in time) integrations. In the BFN method the backward integration of the model resembles the adjoint in the 4DVAR method, but it is less cumbersome to develop. A few studies have shown that BFN compares well with 4DVAR: i) it tends to provide similar accuracy (Auroux and Blum, 2008), and ii) it is less expensive in two ways: the backward integration of the nonlinear model costs less than the adjoint integration, and the method seems to converge in fewer iterations than the 4DVAR. There is a legitimate quest for computationally inexpensive DA methods that account for the time variability in the observations. Continuous data assimilation (CDA) methods fall into this category. Although not being smoothers by nature, CDA methods are computationally inexpensive (because no backward model integration is needed as in the 4DVAR or the BFN methods), and they do account for the time variability in the observations which are continuously assimilated into the forward model as they become available. BFN can however be considered a continuous DA method during the forward integration of the model. Larios and Pei (2018) introduced variations of the CDA method from linear AOT (Azouani, Olson, and Titi, 2013) and applied them to the Kuramoto-Sivashinsky equation (KSE). They showed increasing potential for convergence depending on the form of the model-data relaxation term. The ease of implementation and the potential for convergence of this method makes it attractive for other applications.

This study compares the BFN and Larios and Pei methods using the Lorenz models (Lorenz, 1963, 1996, 2005, 2006; Lorenz and Emanuel, 1998; Baines, 2008). The former has been applied to various models including a complex ocean model (Ruggiero et al., 2015), but it is costly compared to CDA methods. The latter is less expensive, but has not yet been implemented with more complex or chaotic systems to our knowledge. Before attempting an implementation of the Larios and Pei method with a complex ocean model, we first compare its accuracy against the BFN method on three chaotic Lorenz systems. These models provide similar chaos that one would see within an ocean or atmospheric model and have been shown to be an excellent source for evaluating and testing new DA methods (Ngodock et al., 2007). We do note that the KSE is also a chaotic model, but it is not as widely used as a testbed for DA methods as the Lorenz models.





The outline of the paper is as follows. In Section 2, BFN and the Larios and Pei methods are introduced. Section 3 presents the three Lorenz models of increasing complexity used for testing the two methods. Section 4 contains the configuration of the nature run models and the truth for validation, as well as, preliminary testing for the optimal nudging coefficient for each

method. In Section 5, the details of the DA experiments for each model are discussed and results are presented. Lastly, Section 6 contains the conclusion of the experiments.

## 2   Methods

In this section, we discuss the two simpler methods that are compared in this paper. These methods are only briefly presented here, and we refer the reader to the cited references for more details. We note that both the BFN and AOT methods are based

on the well-known nudging algorithm (Hoke and Anthes, 1976).

### 2.1   Diffusive back and forth nudging (D-BFN) method

We start with a simple description of the Back and Forth Nudging (BFN) method proposed by Auroux and Blum (2005, 2008) (Auroux and Nodet, 2011). The BFN method, like nudging, corrects the trajectory as the model is integrated forward in time. The addition in BFN, compared to nudging, is using the state at the end of the assimilation window to initialize the backward

model, which has its own nudging term. It forces the model closer to the observations as it integrates back in time, allowing corrections up to the initial conditions. The adjusted initial condition is then used to initialize the integration of the forward model again and this process is repeated for either a chosen amount of iterations or until a set convergence criterion is reached. Auroux and Blum then introduced the Diffusive Back and Forth Nudging (D-BFN) (Auroux et al., 2011) which has the same underlying methods of BFN but with added control of the diffusive term, allowing a stable backwards integration. The D-BFN

algorithm is described below, using a dynamical model in continuous form:

$$\partial_t X = M(X) + v\Delta X, \quad 0 < t < T, \tag{1}$$

with the initial condition $X(0) = x_0$, where M is the model operator and $v$ is the diffusion coefficient. In reference to their paper, (Auroux et al., 2011), M is used for clarity to represent the model operator instead of F since $F$ is later referenced as the forcing constant. In the dynamical system above, the diffusive term has been separated from the model operator. We leave

the reader with the remark that if there is no diffusion, D-BFN reduces to the original BFN method. The D-BFN method is as follows, for $k \geq 1$,

$$\begin{cases} \partial_t X_k = M(X_k) + v\Delta X_k + K(X_{obs} - H(X_k)), \\ X_k(0) = \tilde{X}_{k-1}(0), \quad 0 < t < T, \end{cases} \tag{2}$$

$$\begin{cases} \partial_t \tilde{X}_k = M(\tilde{X}_k) - v\Delta \tilde{X}_k - K'(X_{obs} - H(\tilde{X}_k)), \\ \tilde{X}_k(T) = X_k(T), \quad T < t < 0, \end{cases} \tag{3}$$


where $X(t)$ is the state vector with initial condition $X(0) = x_0$, $K/K'$ is the feedback or nudging coefficient, and H is the observation operator, which allows comparison of the observations, $X_{obs}$, with the corresponding model state, $H(X(t))$. For D-BFN, as opposed to BFN, the opposite sign of the diffusive coefficient is used to stabilize the backwards model. The nudging coefficients $K$ and $K'$ can have the same or different magnitudes where the equations determine the opposite signs for the nudging terms. For the cases that the non-diffusive portion of the model can be reversed, the backward nudging equation can be rewritten for $t' = T - t$:

$$
\begin{cases}
\partial_{t'} \tilde{X}_k = -M(\tilde{X}_k) + v\Delta \tilde{X}_k + K'(X_{obs} - H(\tilde{X}_k)), \\
\tilde{X}_k(t' = 0) = X_k(T),
\end{cases}
\tag{4}
$$

where the backward model state, $\tilde{X}$, is evaluated at time $t'$. There is a case in which it is reasonable for $K = K'$ and is of interest for geophysical processes. While this slightly different algorithm was implemented and tested, the original D-BFN algorithms were used for the purposes of this paper.

### 2.1.1 Concave-convex nonlinearity (CCN) method

The method being compared, introduced by Larios and Pei (2018), is a modification of the linear AOT method (Azouani et al., 2013). In their paper, they suggest three new nonlinear continuous DA methods. The first approach was a nonlinear adaptation of the linear AOT method. While this method had faster convergence, it retained higher errors for short periods of time. This led them to introduce a hybrid of the two, the Hybrid Linear/Nonlinear Method that strongly corrects deviations for small errors with the nonlinear portion and maintains the linear AOT algorithm for large errors. The success of this method inspired Larios and Pei to take it a step further and exploit the nudging term, proposing the third method, the Concave-Convex Nonlinearity method that also implements nonlinearity on the large errors. This method converged faster and had smaller errors when compared to the previous two methods and AOT. This last method is the one shown below and used for comparison in this paper. It will also be referred to as CCN or $\eta_3(x)$ in the following equations. We start with the same representation of a time continuous model as in Eq. (1), except the diffusive term is no longer required to be separated,

$$\partial_t X = M(X), \quad 0 < t < T.$$

We then add the feedback correction term,

$$\partial_t X = M(X) + \eta(X_{obs} - H(X)),
\tag{5}$$

where the CCN method is as follows for $0 < \gamma < 1$,

$$
\eta(x) = \eta_3(x) := 
\begin{cases}
x|x|^{\gamma}, & |x| > 1, \\
x|x|^{-\gamma}, & 0 < |x| < 1, \\
0, & x = 0.
\end{cases}
\tag{6}
$$

This nonlinear DA method seems straightforward to implement with a high convergence rate and only a forward integration of the model. It is similar to BFN in that it uses a nudging term to correct the model towards the observations during the





integration of the forward model. The results from their paper with the KSE model look promising, but it is important to note that the reference of fast convergence was in comparison to AOT. The CCN algorithm took roughly 17 time units for

convergence, compared to the 50 time units for AOT. These results also used a very dense set of observations, and although the frequency in which observations were brought in is not explicitly stated in their paper, their equations imply that observations are brought in at every timestep. This paper investigates i) if CCN will converge for a shorter time window with a different, more complex model and if the results can still be achieved with sparse observations and ii) if the functional nudging term in CCN is enough to correct the model without the iterations of a backward correction as in BFN.

## 3   Models

This section presents the three models for which the experiments with the proposed methods will be tested. Each of the well-known Lorenz models (Lorenz 63, Lorenz 96, and Lorenz 05) have been consistently used to test new DA methods.

The first model is the three-component Lorenz (1963) model

$$\frac{dx}{dt} = \sigma(y - x)$$
$$\frac{dy}{dt} = x(\rho - z) - y \tag{7}$$
$$\frac{dz}{dt} = xy - \beta z$$

The three components $(x, y, z)$ represent the amplitudes of velocity, the temperature, and the horizontally averaged temperature, respectively (Baines, 2008). The equations also contain three constant parameters that are set to commonly used values known to cause chaos: $\sigma = 10$, $\beta = 8/3$, and $\rho = 28$.

The second model is the Lorenz (1996) model, published in Lorenz (2006) and Lorenz and Emanuel (1998). The Lorenz 96 is a more complex one-dimensional model for the variables or grid points $X_1, \ldots, X_N$. These can be viewed as values of an

unspecified oceanographic quantity such as temperature or salinity. The model equations are

$$\frac{dX_i}{dt} = (X_{i+1} - X_{i-2})X_{i-1} - X_i + F \tag{8}$$

for $i = 1, \ldots, N$ with the constraint of $N \geq 4$ and the assumption of cyclic boundary conditions. In Equation (8), $-X_i$ is the diffusive term, $F$ is the forcing constant set to the value of $8$ to ensure chaotic behavior, and $N = 40$ is a frequently used quantity for the number of variables.

The third model is the Lorenz (2005) model, a one-dimensional model containing grid points, $X_1, \ldots, X_N$, that can also be considered geographical site locations of some general oceanographic measurement. For clarification, $L$ is used in place of $K$ for the model subscripts since $K$ denotes the nudging coefficient in D-BFN. For $N \geq 4$ and a value $L(L \ll N)$, the model equations are

$$\frac{d}{dt}X_n = [X, X]_{L,n} - X_n + F, \tag{9}$$





for $n = 1, \dots, N$, where $[X, X]_{L,n}$ is the advection term defined by

$$[X, Y]_{L,n} = \frac{1}{L^2} \sum_{j=-J}^{J} \sum_{i=-J}^{J} (-X_{n-2L-i} Y_{n-L-j} + X_{n-L+j-i} Y_{n+L+j}). \tag{10}$$

In this model, $-X_n$ is the diffusive term, $F$ is the chosen forcing term, and $L$ is a selected smoothing parameter where $J = L/2$ if $L$ is even or $J = (L-1)/2$ if $L$ is odd. It has the same cyclic boundary conditions as the Lorenz 96 model. The parameters used in this paper are $F = 10$ to cause chaos, $N = 240$ for the number of grid points, and $L = 8$, which is a commonly used value for smoothing. This model can also be rewritten as a summation of weights. For the purposes of this paper, the original equations were implemented.

## 4    Nature run and preliminary results

In this section, we will first discuss how the nature runs are setup for the three models and then show preliminary testing of the models for the optimal nudging coefficient. A nature run is the result from a model being integrated forward without assimilating any data. These model runs without DA are often used to represent the true model nature where portions of the nature run are referred to as the truth. The truth is used to create observations that are assimilated into the model and to evaluate the DA method implemented. Each model starts with a similar setup scheme for the nature run. The nature runs are first initialized with a uniform random distribution between 0 and 1 and integrated forward using a fourth-order Runge Kutta (RK4) time-stepping algorithm (Lambers et al., 2021). While each nature model has a spinup period (i.e., integrates forward in time to create sufficient chaos), the size of the timestep and the length of the spinup are dependent on the models. After an ample amount of time, the current state of the spinup is used as the initial condition for the DA experiments. Specific details for each model are shown in Table 1. The model then continues for another year (72 time units) to produced what is referenced as the nature run. The last four months (24 time units) of the nature run are used as the truth for validation and experiment testing. The Lorenz 05 model is referenced that 1 time unit is approximately 5 days. Since the Lorenz 05 model simplifies to the Lorenz 96 model, it will use the same time units, whereas the Lorenz 63 model is said to be unitless in time.

**Table 1.** Nature run experiment parameters for each model.

| | | Nature Run Setup | | | |
|---|---|---|---|---|---|
| **Model** | **Timestep Algorithm** | **Timestep Size** | **Approx. Time** | **Spinup Time** | **Nature Run** |
| Lorenz 63 | RK4 | $\Delta t = 1/1000$ | 6 minutes | 1 year | 1 year |
| Lorenz 96 | RK4 | $\Delta t = 1/20$ | 6 hours | 1 year | 1 year |
| Lorenz 05 | RK4 | $\Delta t = 1/40$ | 3 hours | 9 years | 1 year |



### 4.1 Lorenz 63 model

The Lorenz 63 model, Eq. (7), is integrated forward with a timestep of approximately 6 minutes (or $\Delta t = 1/1000$ time unit). The model state at the end of the first year of spinup is used as the initial condition for the DA experiments. The truth is shown in Fig. 1(a) and verifies the length of the nature run is acceptable to produce the rotation between the two wings of the

Lorenz attractors. Next, we note that the two initial conditions $(x, y, z)$ are significantly different. The initial condition of truth: $(-12.0355, -15.7630, 26.9678)$ and the initial condition for the DA experiments: $(2.2731, 2.9968, 17.2231)$. Finally, Fig. 1(b) shows the results for each variable $x$ (top), $y$ (middle), and $z$ (bottom) for a two month run with no DA compared to the first two months of truth.

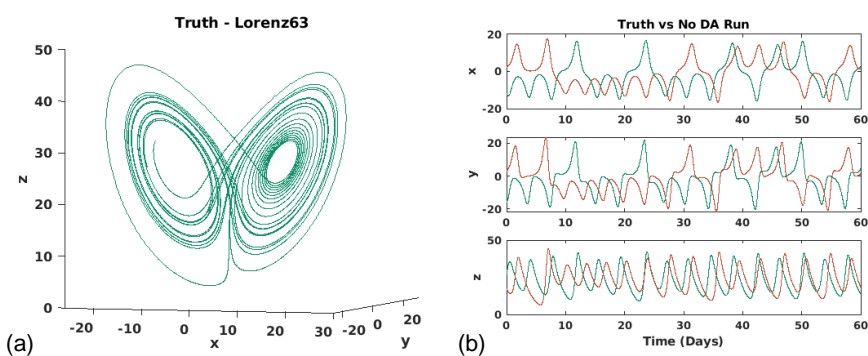

**Figure 1.** (a) Last four months of the nature run referred to as the truth and later used for validation of experiments. (b) The values for each variable $x$ (top), $y$ (middle), and $z$ (bottom) over a two month (12 time units) window. Truth is shown in teal whereas the orange line is a test run with no DA that started with the same background initial condition.

### 4.2 Lorenz 96 model

The Lorenz 96 model, Eq. (8), uses a constant forcing of $F = 8$ and is integrated forward using a 6 hour timestep (or $\Delta t = 1/20$ time unit). The state of the nature run model at the end of the first year of spinup is later used as the initial condition for the DA experiments. Initial conditions for the truth and the background state are shown in Fig. 2(a). Figure 2(b) shows the four months of truth which confirms that the length of spinup and the choice of forcing produced ample chaos. Finally, Fig. 2(c) represents the error between truth and a no DA run. It can be seen that the errors between truth and the no DA run are high and that the

two runs have diverged from each other.

### 4.3 Lorenz 2005 model

The Lorenz 05 model, Eq. (9, 10), uses a constant forcing of $F = 10$ to ensure chaos, an even number $L = 8$, and is integrated forward with a timestep of approximately 3 hours (or $\Delta t = 1/40$ time unit). To produce sufficient chaos, the model spinup integrates forward for 9 years. The initial condition for the DA experiments comes from the model state at the end of the ninth

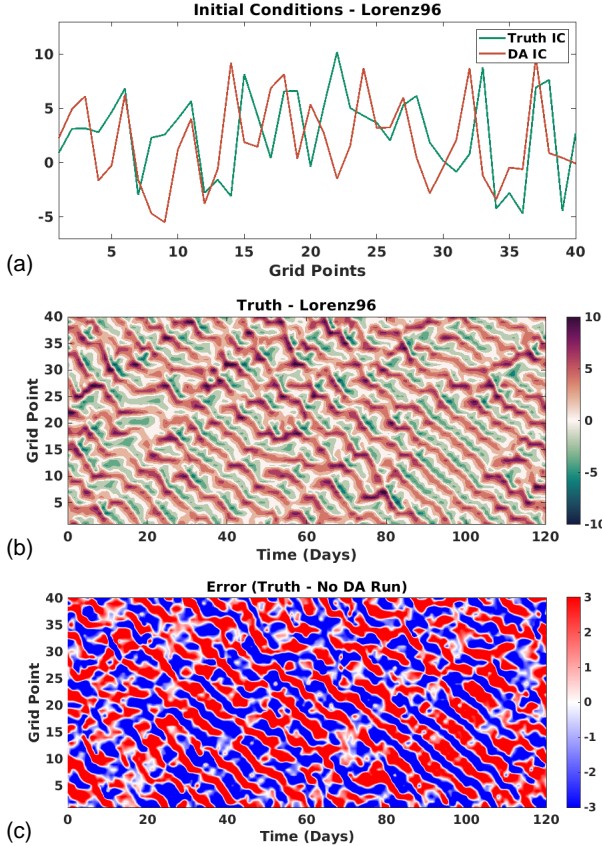

(a)

(b)

(c)

**Figure 2.** These figures capture the Lorenz 96 model setup. The top figure shows the distinction between the initial condition for the truth 'Truth IC' and the initial condition for the DA experiments 'DA IC'. The middle figure is the last four months of the nature run, referred to as the truth and later used for validation of experiments. Lastly, the bottom figure contains the large errors between the truth and a no DA run.

year of the spinup. This initial condition is shown in Fig. 3(a). The model then runs forward another year for the nature run, where the last four months of the nature run, shown in Fig. 3(b), are used as the truth for testing and validation. The initial condition for truth is also shown in comparison in Fig. 3(a). Finally, Fig. 3(c) displays the errors between the truth and a 4-month no DA run.

### 4.4   Preliminary testing

In order to best compare the two methods, we first choose the optimal value for the nudging coefficient for each method and model. The two DA methods, D-BFN and CCN, were implemented for several lengths of time, ranging from 5 days to 2 months. Each DA run was given a set of full observations at all grid points and every timestep. The mean absolute error (MAE, $\frac{1}{N}\sum_{i=1}^{N}|y_i - x_i|$) was computed over time to reflect how well the nudging terms were correcting the model. Several values were chosen for each nudging term: $1 \leq |K| \leq 75$ and $0 < \gamma < 1$.

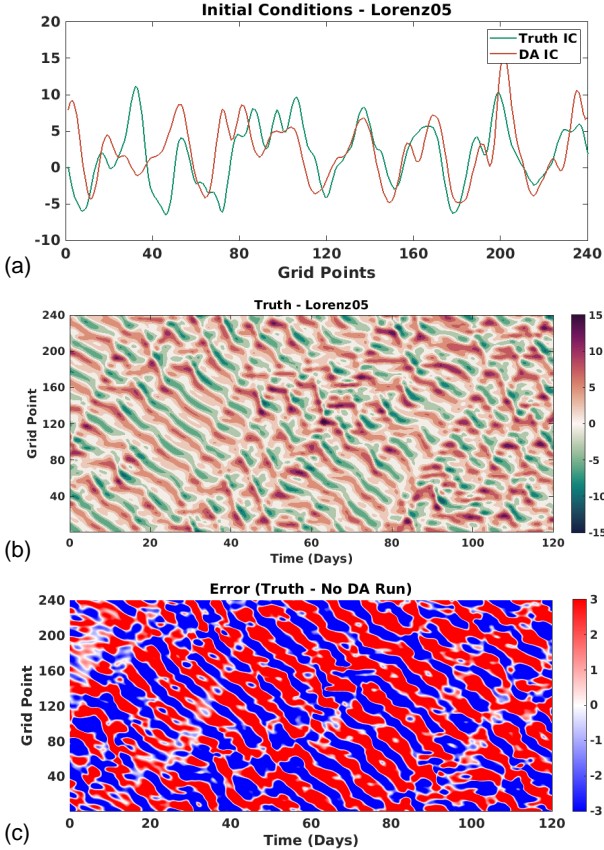

**Figure 3.** These figures capture the Lorenz 05 model setup. The top figure shows the distinction between the initial condition for the truth 'Truth IC' and the initial condition for the DA experiments 'DA IC'. The middle figure is the last four months of the nature run, referred to as the truth and later used for validation of experiments. Lastly, the bottom figure contains the large errors between the truth and a no DA run.

Here, we provide a few remarks. The first is that the "best choice" for the value chosen can be different depending on the model being used. There are other cases discussed in the results section below where the optimal value had to be changed to adapt to the parameters given. Secondly, notice the time length used in the figures, especially for CCN, as it was shown in the original paper (Larios and Pei, 2018) that it takes time to converge. It will also have a higher starting MAE since CCN only corrects the model in the forward integration. Lastly, only a few examples of the preliminary testing are shown in Fig. 4 as not to cloud the paper with repetitive figures.

The first set of figures, Fig. 4(a) and Fig. 4(b), show the preliminary results for testing within a 5 day window for several values of the nudging coefficients. The second set of figures, Fig. 4(c) and Fig. 4(d), display preliminary results for testing within a 30 day window and are added to show that CCN converges when given a sufficient amount of time. In this case, CCN converges around 10 days for the largest nudging coefficient.


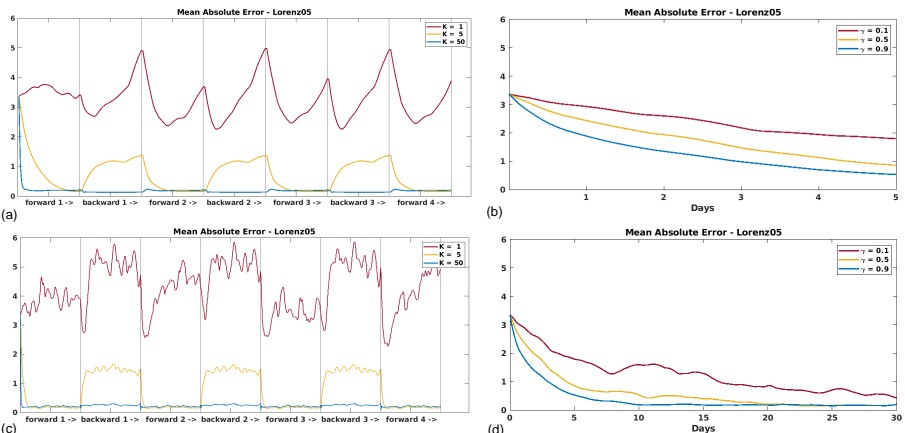

**Figure 4.** (a,c) D-BFN (b,d) CCN. Preliminary testing to choose the optimal value for the nudging coefficients for each method. Results shown are for the Lorenz 05 model with a 5 day and a 30 day (1 month) data assimilation window. Observations are brought in at every grid point and timestep (timestep for Lorenz 05 model is 3 hours with 240 grid points).

## 5 Data assimilation experiments: Setup and results

Several experiments are carried out with different lengths of DA windows. The length of the forecast is the same as the time window chosen for DA. The observations for these experiments will come from the start of truth for the DA window length chosen. For example, an assimilation length of two months will bring in observations from the first two months of truth and the DA accuracy is compared to these first two months. The latter two months of truth are then used for testing the two month forecast accuracy.

### 5.1 Lorenz 63 model

The first set of experiments is carried out with the three-component Lorenz 63 model, Eq. (7). All experiments have the same parameters of $\sigma = 10, \beta = \frac{8}{3}$, and $\rho = 28$ with a timestep of approximately six minutes ($\Delta t = 1/100$). Preliminary testing was done to choose the best value for the nudging terms. For CCN, the best value of $\gamma$ is 0.9 and for D-BFN, any value of $K = 25$ or larger provided accurate results. The value of $K = 25$ was chosen because there was no improvement in the accuracy of the DA experiments with higher values of $K$.

The experiments started with shorter time windows of 5 and 10 days of DA along with a 5 and 10 day forecast, respectively. For the best results possible, observations were brought in at all grid points and every timestep. Table 2 shows the mean absolute error (MAE) averaged over time for both methods (D-BFN and CCN) and for the DA run and the forecast (FC). While D-BFN does well with a short time window, CCN does not have an adequate amount of time for corrections to make an impact on the DA error. The time window was then lengthened to a one and two months DA run along with a one and two month forecast. The results are shown in Table 2, as well as Fig. 5. While CCN shows higher MAE for not having a long enough time window to reduce errors, the forecast MAE is on par with D-BFN for the one month forecast and slightly better than D-BFN for the two





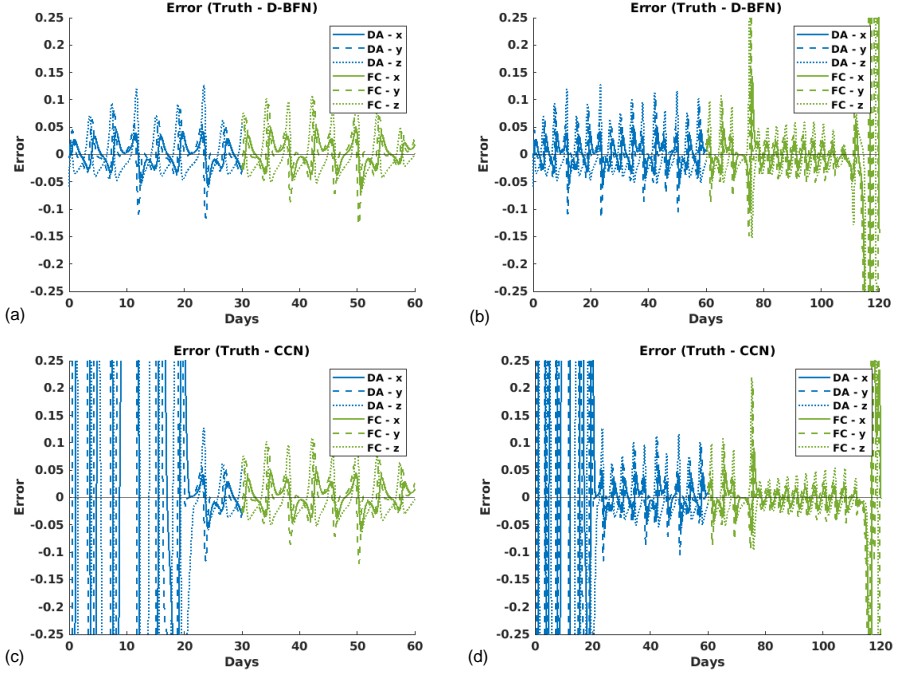

**Figure 5.** (a,b) D-BFN ($K = 25$) and (c,d) CCN ($\gamma = 0.9$). Error plots between experiments with the Lorenz 63 model compared with the truth for each variable, x (solid line), y (dashed line), z (dotted line). DA is shown in blue and the forecast (FC) is shown in green. The left column shows experiments with 1 month of DA and 1 month of forecast. The right column shows experiments with 2 months of DA and 2 months of forecast. All DA experiments assimilated all observations (i.e., all grid points at every timestep/6 minutes).

month forecast. Figures 5(c) and 5(d) show that CCN has better accuracy in the forecast for several days longer than D-BFN

when given sufficient time to make corrections.

Further experiments were done in the case when all observations are not available. The experiment '*1gp2ts*' shown in Table 2 is bringing in observations at all grid points but now every other timestep. These were only performed for the longer time windows of one and two months. D-BFN still provided high accuracy with less observations in time but CCN was not able to make a suitable correction within this time window. Several factors play a role in this outcome starting with internal factors

of D-BFN, namely the backwards integration of the model and the iterations. The backwards integration helps propagate the correction from the nudging term further into the model domain, an ability that is not present in CCN. It can also been seen in Fig. 4 that the rate in which corrections are made imply that D-BFN has a stronger nudging term compared to CCN. It is possible that if a longer time window were considered, CCN would produce lower errors for the DA run and the forecast. It was shown in the original paper that it took approximately 17 time units to converge with the KSE model, and these experiments

are 6 time units (1 month) and 12 time units (2 months).

Other experiments were tested that are not shown in this paper but should be discussed. For all of the window lengths used for this model (5 days, 10 days, 1 month, and 2 months), observations were brought in for all variables every five timesteps,





**Table 2.** Table of DA experiments. Observations used: *'all obs'* = all observations (every 6 minutes), *'1gp2ts'* = all grid points, every other timestep (every 12 minutes). *'5d'*, *'10d'*, *'1m'*, and *'2m'* represent 5 days, 10 days, 1 month, and 2 months, respectively. *'DA'* is the window for data assimilation and *'Fcast'* is the forecast window. Values shown are the time averaged MAE.

| | | Lorenz 63 Model | | | |
|---|---|---|---|---|---|
| **Observations** | **DA Method** | **5d DA** | **5d Fcast** | **10d DA** | **10d Fcast** |
| **all obs** | D-BFN ($K = 25$) | 0.0221 | 0.0225 | 0.0224 | 0.0279 |
| | CCN ($\gamma = 0.9$) | 3.1663 | 3.3081 | 2.2818 | 6.8962 |
| | | **1m DA** | **1m Fcast** | **2m DA** | **2m Fcast** |
| **all obs** | D-BFN ($K = 25$) | 0.0247 | 0.0254 | 0.0254 | 0.1766 |
| | CCN ($\gamma = 0.9$) | 1.2605 | 0.0256 | 0.6434 | 0.0591 |
| **1gp2ts** | D-BFN ($K = 25$) | 0.0317 | 0.0255 | 0.0508 | 0.0924 |
| | CCN ($\gamma = 0.9$) | 7.9482 | 6.4443 | 8.0473 | 10.2191 |

approximately, every 30 minutes. While it is not unpredictable that CCN did not do well with even less observations, D-BFN was still able to produce good results. We present this experiment for the example of how the nudging coefficient needs

to sometimes be adjusted. The value of $K = 25$ produced high forecast accuracy for the shorter time windows but did not converge for the longer ones. Increasing the value to $K = 50$ produced similar results as $K = 25$ for the shorter windows but also produced accurate results for the longer time windows. The conclusion from these results was that a larger nudging coefficient was needed for D-BFN in cases with sparse observations and/or longer time windows.

The results above confirmed that a longer time window is still needed with these models in order for CCN to converge.

Therefore, the next two models will have only the longer DA runs. For these experiments, the two lengths of assimilation and forecasting considered are one and two months followed by their respective forecast.

## 5.2   Lorenz 96 model

All numerical experiments for Lorenz 96, Eq. (8), will use the following parameters: $N = 40$ grid points, $F = 8$, and a time step of approximately 6 hours ($\Delta t = 1/20$). The preliminary testing revealed the best choice of $K = 25$ and $\gamma = 0.9$ for D-BFN

and CCN, respectively.

The first set of experiments with this model use observations at all grid points and all timesteps. The averaged Mean Averaged Error (MAE) is shown in Table 3 where CCN produces a slightly better forecast than D-BFN. Of course, CCN has a higher error for DA since it only corrects in the forward model. Figures 6(a) and 6(c) for the one month experiment show how long the forecast is accurate, which is around 12–15 days for both methods. Figures 6(c) and 6(d) have the results for the two month

experiment showing that the accuracy in forecast for D-BFN has dropped to around 5 days, whereas, CCN is consistent with accuracy for about 12–15 days.


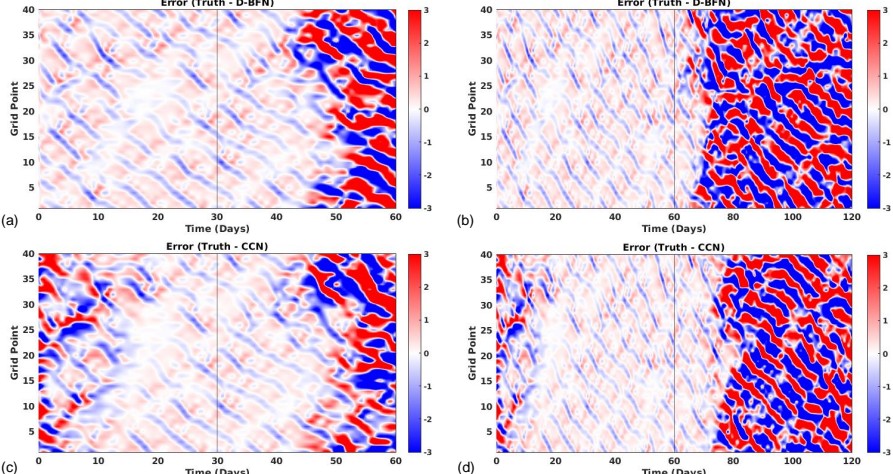

**Figure 6.** (a,b) D-BFN ($K = 25$) and (c,d) CCN ($\gamma = 0.9$). Error plots between experiments with the Lorenz 96 model compared with the Truth. The left column shows experiments with 1 month of DA and 1 month of forecast. The right column shows experiments with 2 months of DA and 2 months of forecast. The vertical line represents the change from the DA window to the forecast window. All DA experiments *'allobs'* assimilated all observations (i.e., all grid points every timestep/6 hours).

The next set of experiments brought in observations at all grid points and every other timestep *'1gp2ts'*. Figure 7 shows the error between truth and each method along with their forecast. D-BFN produces similar results as compared to assimilating all observations. CCN, however, does not make much of a correction during assimilation, which in return does not produce a usable forecast. We would hypothesize that CCN needs a much longer assimilation window to account for not having a full observation set. We carried out experiments with smaller and slightly higher values for $\gamma$, but the resulting assimilation and forecast errors did not improve. (Results not shown).

**Table 3.** Table of DA experiments. Observations used: *'all obs'* = all observations (every 6 hours), *'1gp2ts'* = all grid points, every other timestep (every 12 hours). *'1m'* and *'2m'* represent 1 month and 2 months, respectively. *'DA'* is the window for data assimilation and *'Fcast'* is the forecast window. Values shown are the time averaged MAE.

| | **Lorenz 96 Model** | | | | |
|---|---|---|---|---|---|
| **Observations** | **DA Method** | **1m DA** | **1m Fcast** | **2m DA** | **2m Fcast** |
| **all obs** | D-BFN ($K = 25$) | 0.4006 | 1.8820 | 0.4036 | 3.6572 |
| | CCN ($\gamma = 0.9$) | 0.7620 | 1.5284 | 0.5581 | 3.1434 |
| **1gp2ts** | D-BFN ($K = 25$) | 0.4062 | 1.8197 | 0.4075 | 3.4985 |
| | CCN ($\gamma = 0.9$) | 1.9662 | 3.6858 | 1.6443 | 3.8755 |


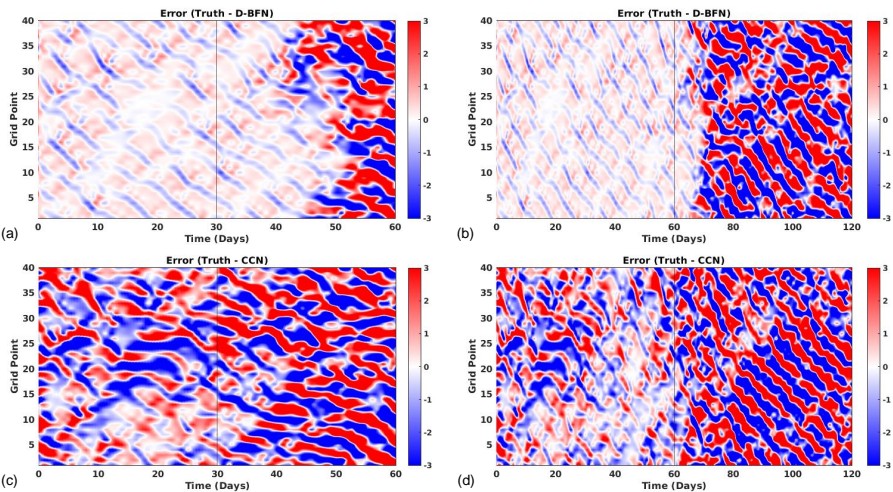

**Figure 7.** (a,b) D-BFN ($K = 25$) and (c,d) CCN ($\gamma = 0.9$). Similar to Fig. 6 except the DA experiments *'1gp2ts'* assimilated observations at all grid points and every other timestep (every 12 hours).

**Table 4.** A variety of other experiments testing the sparsity of observations. The first number represents how many spaces between grid points whereas the second represents the time between timesteps. For example, *'3gp2ts'* are observations brought in at every three gridpoints and every two timesteps. Recall that one timestep is equal to 6 hours for this model, so every two timesteps would be every 12 hours. Values shown are the time averaged MAE.

| | **Lorenz 96 Model** | | |
| --- | --- | --- | --- |
| **Observations** | **DA Method** | **1m DA** | **1m Fcast** |
| 1gp5ts | D-BFN ($K = 25$) | 0.4255 | 3.2234 |
| | CCN ($\gamma = 0.9$) | 2.6634 | 4.1855 |
| 1gp10ts | D-BFN ($K = 25$) | 0.5181 | 3.5457 |
| | CCN ($\gamma = 0.9$) | 3.0572 | 4.2346 |
| 1gp20ts | D-BFN ($K = 25$) | 1.8630 | 4.1870 |
| | CCN ($\gamma = 0.9$) | 3.7072 | 4.2537 |
| 2gp2ts | D-BFN ($K = 25$) | 0.9046 | 3.7016 |
| | CCN ($\gamma = 0.9$) | 2.6375 | 4.1872 |
| 3gp2ts | D-BFN ($K = 25$) | 1.7059 | 3.9207 |
| | CCN ($\gamma = 0.9$) | 3.0278 | 4.3588 |
| 4gp3ts | D-BFN ($K = 25$) | 2.1865 | 3.9240 |
| | CCN ($\gamma = 0.9$) | 3.3608 | 4.0710 |

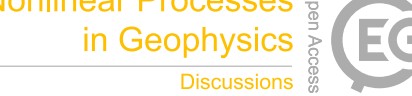

A few other experiments were performed to test the capabilities of these methods with sparse observations. All of these were completed with the two month DA window. Observations were assimilated less frequently in time, from every five *'1gp5ts'* to

every ten *'1gp10ts'* to every twenty *'1gp20ts'* timesteps. The results are displayed in Table 4. The results for CCN are poor as it did not have enough observations to make a correction in the forward model. D-BFN has the benefit of propagating the observations back in time, correcting the initial conditions, and running the forward model again. This process allows D-BFN to give a much better correction during the assimilation window. However, the forecast accuracy decreases with the frequency of observations. The results for every five timesteps (every 30 hours) is comparable to the results from all observations. The

days of accuracy for the less frequent observations drastically decrease as the observations decrease.

## 5.3 Lorenz 2005 model

The Lorenz 05 model, Eq. (9) and Eq. (10), will use the same parameters for all numerical experiments: 240 grid points ($N$), an even number $L = 8$, a forcing constant of 15 to ensure chaos ($F$), and a time step of approximately 3 hours ($\Delta t = 1/40$ time unit). Recall that in Eq. (9) and Eq. (10), one unit of time is equivalent to 5 days. In the preliminary testing, shown in Fig.

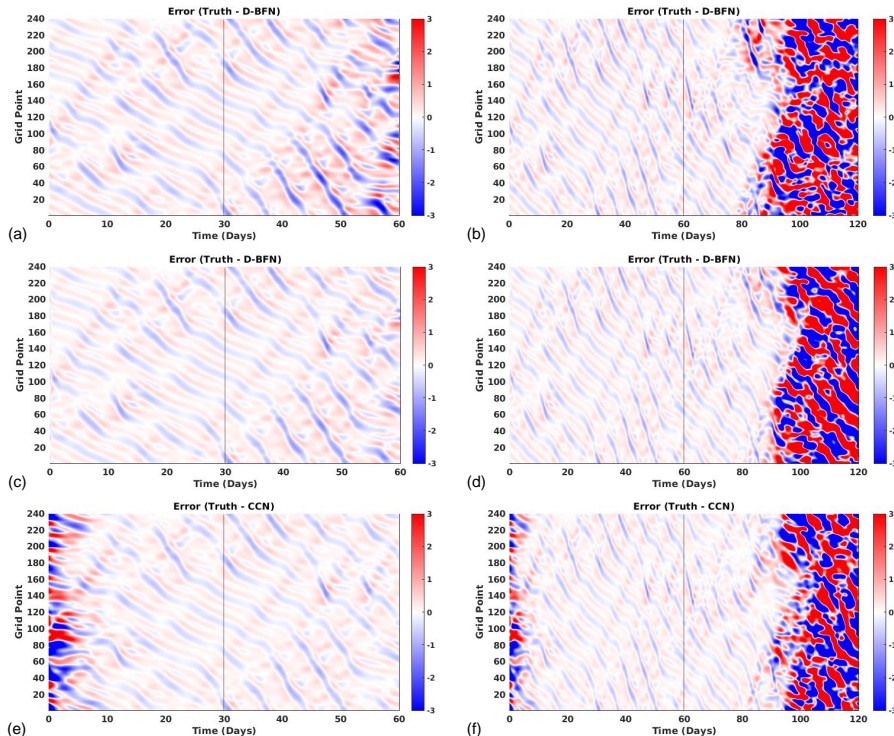

**Figure 8.** (a,b) D-BFN ($K = 50$), (c,d) D-BFN ($K = 25$), and (e,f) CCN ($\gamma = 0.9$). Error plots between experiments with the Lorenz 05 model compared with the Truth. The left column shows experiments with 1 month of DA and 1 month of forecast. The right column shows experiments with 2 months of DA and 2 months of forecast. The vertical line represents the change from the DA window to the forecast window. All DA experiments 'allobs' assimilated observations at all grid points and every timestep (i.e. every 3 hours).



4, the values tested for D-BFN were $K = 1, 5, 50$. The quickest convergence was shown with $K = 50$. Other values were used

as well, such as $K = 25$, which had very similar convergence to $K = 50$. Due to them having nearly identically results, both

values are tested in the original experiments. Consistent with the previous models, the optimal value found for CCN is $\gamma = 0.9$.

The first set of experiments with this model use observations at all times and space for one and two months DA. Two values

for the nudging coefficient $K$ are used to further evaluate the best choice. The results are quite close for $K = 50$ and $K = 25$,

but $K = 25$ has a lower error at the end of the DA window and a slightly better forecast. For this model, CCN has the lowest

forecast accuracy of all results for both the one month and two month. The forecast has low errors for around 30 days, as seen

in Fig. 8.

The second set of experiments uses all points in space and assimilates them at every other timestep. D-BFN produces very

similar results as with the all observations experiment. Looking at the difference in results between one month and two month,

the CCN method needs a longer window to converge with the sparser set of observations, as seen in Fig. 9. Table 5 contains

further details of the mean absolute errors for the first two sets of experiments. The values in Table 5 are separated to show

error contained during the DA window and error maintained during the forecast window.

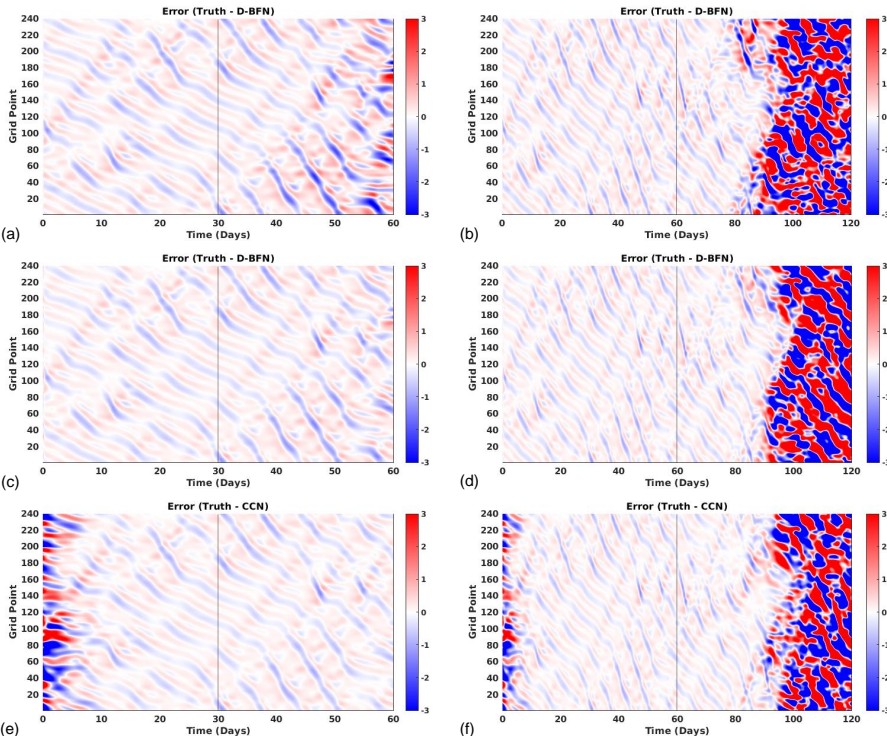

**Figure 9.** (a,b) D-BFN ($K = 50$), (c,d) D-BFN ($K = 25$), and (e,f) CCN ($\gamma = 0.9$). Similar to Fig. 8 except the DA experiments *'1gp2ts'* assimilated observations at all grid points and every other timestep (every 6 hours).





**Table 5.** Table of DA experiments. Observations used: *'all obs'* = all observations (every 3 hours), *'1gp2ts'* = all grid points, every other timestep (every 6 hours). *'1m'* and *'2m'* represent 1 month and 2 months, respectively. *'DA'* is the window for data assimilation and *'Fcast'* is the forecast window. Values shown are the time averaged MAE.

| | **Lorenz 05 Model** | | | | |
|---|---|---|---|---|---|
| **Observations** | **DA Method** | **1m DA** | **1m Fcast** | **2m DA** | **2m Fcast** |
| **all obs** | D-BFN ($K = 50$) | 0.2095 | 0.3856 | 0.2959 | 1.8137 |
| | D-BFN ($K = 25$) | 0.1827 | 0.2480 | 0.1960 | 2.1770 |
| | CCN ($\gamma = 0.9$) | 0.3984 | 0.1948 | 0.2246 | 2.1161 |
| **1gp2ts** | D-BFN ($K = 50$) | 0.2102 | 0.3718 | 0.2249 | 2.2100 |
| | D-BFN ($K = 25$) | 0.1861 | 0.2417 | 0.1977 | 1.9577 |
| | CCN ($\gamma = 0.9$) | 0.8913 | 2.9029 | 0.5941 | 3.3410 |

**Table 6.** A variety of other experiments testing the sparsity of observations. The first number represents how many spaces between grid points whereas the second represents the time between timesteps. For example, *'3gp2ts'* are observations brought in at every three gridpoints and every two timesteps. Recall that one timestep is equal to 6 hours for this model, so every two timesteps would be every 12 hours. Values shown are the time averaged MAE.

| | **Lorenz 05 Model** | | |
|---|---|---|---|
| **Observations** | **DA Method** | **1m DA** | **1m Fcast** |
| **1gp5ts** | D-BFN ($K = 25$) | 0.2095 | 1.5355 |
| | CCN ($\gamma = 0.9$) | 2.2238 | 4.5812 |
| **1gp20ts** | D-BFN ($K = 25$) | 0.3997 | 1.9565 |
| | CCN ($\gamma = 0.9$) | 3.5654 | 4.3697 |
| **2gp2ts** | D-BFN ($K = 25$) | 0.6533 | 3.7837 |
| | CCN ($\gamma = 0.9$) | 2.3233 | 4.3569 |
| **3gp2ts** | D-BFN ($K = 25$) | 1.0572 | 3.9058 |
| | CCN ($\gamma = 0.9$) | 2.5416 | 4.3568 |
| **4gp3ts** | D-BFN ($K = 25$) | 2.0827 | 4.2232 |
| | CCN ($\gamma = 0.9$) | 3.6660 | 4.7131 |

D-BFN does well compared to CCN for observations that are sparse in time. Table 6 shows the results for a two month DA and forecast for observations brought in every five *'1gp5ts'* and every twenty *'1gp20ts'* timesteps. The correction in the DA brings the error down to provide a decent forecast. The error in the forecast is relatively low compared to the errors in CCN
and the larger errors are towards the end of the forecast period. The figure is not shown in this paper but both results have high accuracy for approximately the first 30 days of the forecast.

## 6 Conclusions

Overall each method has their own advantages and disadvantages. While D-BFN performs better with short windows and sparse observations, it does require iterations of forward and backward integrations of the model. This is not suitable for all
cases, most importantly when a model cannot be integrated backwards. For some cases where the assimilation window was long enough, the DA error at the end of the window was lower from the CCN method than D-BFN, resulting in a forecast that maintained accuracy longer in time. Furthermore, CCN only requires the forward model, which is useful for models that do not allow a backwards integration and also makes this method more computationally efficient.

We want to remember a goal of this paper was to determine the best method to apply to an ocean model. For this reason,
we do not want to implement a longer time window as it is not practical for ocean DA. In terms of implementing either method for an ocean model, based on the findings in this paper, Auroux and Blum's D-BFN method seems more applicable to the assimilation window constraints and sparse ocean observations available. However, the implementation of CCN may be suitable for other scenarios with a long assimilation in the ocean such as done in reanalysis or assimilations that start much further in the past.
The results from this paper led us to the conclusions above, but we leave the reader with this final remark. While D-BFN is able to retain accuracy for observations that are sparse in time, due to the advantage of spreading these corrections through the back and forth iterations, we observed that the results from CCN decayed as the density and/or frequency of observations were reduced. These results may be partial to the models not having strong dynamics capable of propagating the corrections to other unobserved points in space or time. However, for models with strong advection, the corrected term may be able to
disperse these corrections to places where observations are not observed, which would allow CCN to have a higher impact when adjusting the trajectory from sparse observations.

*Author contributions.* Vivian A. Montiforte developed the code for the models and methods, performed the experiments, and prepared the manuscript with contributions from co-authors. Hans E. Ngodock proposed the research topic, provided knowledge and background, and offered mentorship throughout this research. Innocent Souopgui provided knowledge and assistance during code development.

*Competing interests.* The authors declare that they have no conflict of interest.

*Acknowledgements.* Vivian A. Montiforte was supported by the U.S. Naval Research Laboratory (NRL) through a postdoctoral fellowship with the American Society for Engineering Education (ASEE). The authors thank numerous NRL colleagues for their collaborative thinking





and instructive input throughout experimentation and editing of this paper, specifically John J. Osborne who offered guidance during the initial implementation of BFN.





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
