# Peer review of "A Comparison of Two Nonlinear Data Assimilation Methods"

_Nonlinear Processes in Geophysics, 2024_

## Author Response (AR2)

**Reviewer/Author Comments and Manuscript Revisions**

**Reviewer Comments 1 (RC1)**

*General Comments*

   *The authors provide a set of methods that seek to provide some time dependence to their fit to observations, like 4D-Var, but with a potentially lower computational burden.*

   *The introduction is a bit imprecise, and would benefit from an extended literature review on the concepts and categorizations of DA and filtering/smoothing methods. Further, the works from Wang, Pu, Kalnay, etc. (citations below) should all be discussed in the context of alternative methods since they were proposed over 20 years ago and seem to have relevance to the D-BFN method and have been tested in a more operational context. There is a lot of historical work from the mathematical field "synchronization of chaos" using Lorenz 63 in particular, but also more recently with Lorenz-96 and the shallow water equations. A starting point would be to look at the works of Pecora and Carroll, as well as Abarbanel and coauthors. I suggest the authors review these works and put some of their discussion in context of these efforts, which largely used nudging methods to achieve synchronization and are highly related to DA methods.*

   *It would be very useful to apply a conventional DA method as a baseline for comparison - e.g. either 3D-Var or 4D-Var would be good choices. 4D-Var would be ideal since the authors propose the methods here as alternatives to 4D-Var, but at least a simple 3D-Var (with a reasonably well calibrated background error covariance matrix B) could serve as a good control. Ideally, the authors would provide both and show where these new methods fall in relation to those. There is also not really any discussion of the costs of the proposed methods in comparison to 4D-Var and 3D-Var, though this is included as advantage of these methods in the manuscript text, and it would be helpful to put the proposed methods in that context for potential consideration for future research and development.*

   *The description of the experiment setup should be more precise. The details are unclear and require guessing on the part of the reader. In addition, it would be useful to have a section describing the observation sampling strategy, the density/sparsity of the observations, and the noise applied to the observations, prior to describing the DA setup and results.*

   *More investigation would be useful to show how robust the methods are to sparsity in observations in both space and in time, and with increasing noise in the observations. Further, the discussion and results should be further separated between a spinup period (i.e. what conditions are required to have the different DA methods drive the state estimate toward the true state), the "online DA" period (i.e. once the system is spun up, can it maintain the state estimates with reasonable accuracy without filter divergence), and the forecast. At the moment these are all combined into single experiments, and so make it difficult to establish the behavior of the methods. For example, after spinup, long forecasts can be initialized from every DA analysis initial condition in order to produce forecast statistics - as opposed to a single forecast at the end of a cycled DA period. This would be more consistent with how operational forecasts are conducted. Overall, additional efforts like this are needed to improve the statistical reliability of the experiment results reported.*

   **Author's Comment (AC):** The introduction has been revised, and the methods are referred to as sequential and non-sequential, instead of filters and smoothers.

   We have not included a conventional baseline comparison within this manuscript because it has previously been tested and results published. See reference below containing the results between the BFN and 4DVAR.
   *Auroux, D. and Blum, J.: A nudging-based data assimilation method: The back and forth nudging (BFN) algorithm, Nonlin. Proc. Geophys., 15, 305–319, https://doi.org/10.5194/npg-15-305-2008, 2008.*

   We have revised the explanation of the model setup and added a visual diagram to aid in helping the reader understand the model setup. Further details in comment response to L170.

   There was some confusion about the results containing the spinup. We have clarified in replies to specific comments that the spinup is only used to initialize the models before starting experiments. It is not included in any experiment results.

*Manuscript Change (MC)*: Line numbers are referenced from updated manuscript.

**Specific/Technical Comments**

LINE 5, *"The second is the Concave-Convex Nonlinearity (CCN) method provided by Larios and Pei that has a straightforward implementation and promising results."*

**RC1**: *Promising results in what context? Toy models? Full scale ocean models?*

**AC**: "promising results with a toy model". The promising results being referred to in their paper used the Kuramoto-Sivashinsky equations. This model is not stated in the abstract but in the introduction L45-46. "Larios and Pei (2018) introduced variations of the CDA method… and applied them to the Kuramato-Sivashinsky equation.

**MC:** See Line 7.

LINE 8, *"integrating the model forward and backward in time"*

**RC1**: *It might be worth mentioning whether this means with the TLM and Adjoint, as done with 4D-Var, or it requires a different model to describe the "backward in time" integration.*

**AC**: No, this is not done with the TLM and adjoint. Although it may resemble the adjoint, the backward integration is done with the nonlinear model, as described in Auroux, Blum, & Nodet (2011), which also accounts for the diffusive processes. This sentence has been revised in the manuscript.

> Auroux, D., Blum, J., and Nodet, M.: Diffusive back and forth nudging algorithm for data assimilation, C. R. Math., 349, 849–854, https://doi.org/10.1016/j.crma.2011.07.004, 2011.

**MC:** See Lines 8-9. Revised the sentence.

LINE 11, *"is fairly adaptive to sparser observations, predominately in time."*

**RC1**: *I think you mean 'robust' not 'adaptive': "is fairly robust to sparser observations, predominately in time."*

**AC**: We have adopted the suggestion from the reviewer and have revised the text.

**MC:** See Line 12.

LINE 14, *"There are generally two classes of data assimilation (DA) methods: filters and smoothers."*

**RC1**: *I'm not sure that I agree with this characterization. Sequential DA methods can be applied as smoothers - for example the ensemble Kalman filter can be applied as a smoother to update states throughout an analysis window. It is more typical to separate the classes of conventional DA methods into 'sequential' and 'variational'.*

**AC***:* In the revised manuscript we have adopted the characterization of DA methods into sequential and non-sequential instead of filters and smoothers.

**MC:** See Lines 15-18, as well as Introduction.

LINE 19, *"Filters assume that all the data within the observation window are collected and valid at the analysis time."*

**RC1**: *I don't think this is the key distinguishing factor between a filter and a smoother.*

*For example, from Doucet and Johansen (2008):*
> *https://www.stats.ox.ac.uk/~doucet/doucet_johansen_tutorialPF2011.pdf*

*"filtering methods have become a very popular class of algorithms to solve these estimation problems numerically in an online manner, i.e. recursively as observations become available,"*

*They define filtering as:*
> *"the problem of filtering: characterising the distribution of the state of the hidden Markov model at the present time, given the information provided by all of the observations received up to the present time"*

*While they define smoothing as:*
> *"smoothing corresponds to estimating the distribution of the state at a particular time given all of the observations up to some later time"*

*In that sense, even 3D-Var is a smoother if observations are used throughout a window and the analysis is formed at the center of that window (a common operation at NOAA, for example).*

**AC**: We refer to the reply to the previous comment. We are no longer discussing filters and smoothers, rather sequential and non-sequential DA methods.

**MC:** See Lines 15-18, as well as Introduction.

**LINE 23**, *"Smoothers on the other hand assimilate all observations collected within the observation window at their respective time and provide a correction to the entire model trajectory over the assimilation window."*

**RC1**: *Again, this is not specific to a smoother. I suggest the authors just remove the filter/smoother distinction and focus on the key property that observations are assimilated throughout a time window (which all modern operational forecast systems do at this stage, either using 3D-Var FGAT, a 4D-EnKF, or 4D-Var).*

**AC**: The filter/smoother characterization has been replaced with sequential and non-sequential.

**MC:** See Lines 15-18, as well as Introduction.

**LINE 25-26**, *"The former refers to the time window over which a correction to the model is computed, while the latter refers to the time window over which observations are collected/considered for assimilation."*

**RC1**: *I don't know of many systems that don't have these two windows coincide, but I am aware that the SODA system at UMD uses a longer observation window for each analysis, which might be worth citing here as an example ocean DA system that follows this strategy.*

**AC**: In addition to the SODA system, a reference to the Navy coupled ocean data assimilation system (NCODA, Cummings, 2005) has been added to this paragraph in the revised manuscript.
> *Carton, J. A., G. Chepurin, X. Cao, and B. Giese, 2000: A Simple Ocean Data Assimilation Analysis of the Global Upper Ocean 1950–95. Part I: Methodology. J. Phys. Oceanogr., 30, 294–309, https://doi.org/10.1175/1520-0485(2000)030<0294:ASODAA>2.0.CO;2.*

> *Cummings, J. A.: Operational multivariate ocean data assimilation, Quart. J. Roy. Meteorol. Soc., 131, 3583‑3604, https://doi.org/10.1256/qj.05.105, 2005.*

**MC:** See Lines 31-33. Added the two references above.

**LINE 29-31**, *"There are a few known smoother methods such as the four-dimensional variational (4DVAR) (Fairbairn et al., 2013; 30 Le Dimet and Talagrand, 1986), the Kalman Smoother (KS) (Bennett and Budgell, 1989), and the Ensemble Kalman Smoother (EnKS) (Evensen and Van Leeuwen, 2000). Of these three, 4DVAR is the one that is most used in geosciences problems."*

**RC1**: *By your definition, the 4D Local Ensemble Transform Kalman Filter (LETKF) is also a smoother. I'd argue that the EnKF is potentially used more in geoscience problems due to its ease of implementation compared to 4D-Var. So I would just say: "Of these three, 4DVAR is considered one of the leading state-of-the-art methods for geosciences problems."*

**AC**: We agree with the reviewer and have adopted the suggestion.

**MC:** See Line 39.

LINE 32-33, *"[4DVAR] **does, however, require the development of a tangent linear (TLM) and adjoint of the dynamical model being used. This development of the TLM and the adjoint model is both cumbersome and tedious,** [and requires regular maintenance as the base model undergoes continued development]**."***

**AC**: We agree with the reviewer and have adopted the suggestion.

**MC:** See Line 41.

LINE 38-39, "the backward integration of the nonlinear model costs less than the adjoint integration"

**RC1**: *How exactly do you propose to integrate a model backwards in time, particularly one that has diffusive processes?*

**AC**: The backwards integration of the model was described in Auroux et al (2011) when they introduced the Diffusive Back and Forth Nudging method.

> *Auroux, D., Blum, J., and Nodet, M.: Diffusive back and forth nudging algorithm for data assimilation, C. R. Math., 349, 849–854, https://doi.org/10.1016/j.crma.2011.07.004, 2011.*

**RC1**: *Over 20 years ago, Kalnay et al. (2000) proposed the 'quasi-inverse' method to do something similar. While they made parallels to 3D-Var, it was actually similar to what is being described here as an alternative to 4D-Var. I think it would be worthwhile to compare the quasi-inverse method since they faced similar challenges with reverse-propagation of the nonlinear system, and gave an example applying this in the NOAA operational forecast system (e.g. running the TLM backwards with the sign of surface friction and horizontal diffusion changed).*

> *Kalnay, E., S. K. Park, Z. Pu, and J. Gao, 2000: Application of the Quasi-Inverse Method to Data Assimilation. Mon. Wea. Rev., 128, 864–875, https://doi.org/10.1175/1520-0493(2000)128<0864:AOTQIM>2.0.CO;2.*

This built on the work of Wang et al. (1995) and Pu et al. (1997a/b)

> *Wang, Z., I. M. Navon, X. Zou, and F. X. Le Dimet, 1995: A truncated Newton optimization algorithm in meteorology applications with analytic Hessian/vector products. Comput. Optim. Appl.,4, 241–262.*
>
> *Pu, Z.-X., E. Kalnay, J. Sela, and I. Szunyogh, 1997a: Sensitivity of forecast errors to initial conditions with a quasi-inverse linear model. Mon. Wea. Rev.,125, 2479–2503.*
>
> *Pu, Z.-X., E. Kalnay, J. Derber, and J. Sela, 1997b: An inexpensive technique for using past forecast errors to improve future forecast skill. Quart. J. Roy. Meteor. Soc.,123, 1035–1054.*

**AC**: We do not see the need to make a parallel with the quasi-inverse. The Back-and-Forth method has already been described in many publications, some cited in the manuscript.

LINE 80-90, EQUATIONS 2,3,4,

**RC1**: *It seems that K and K' are constants here. Much of the work in modern DA is formulating K as a matrix operator - for example the Kalman gain matrix. In that case, K contains all of the information about the forecast error, observation error, and potentially model error.*

**AC**: The reviewer is right that a better choice for K is the Kalman gain matrix. However, the use of K as the Kalman gain matrix implies more computational expenses (matrix inversion), which we are attempting to avoid with the simpler methods implemented in this study. However, using K as a matrix with the BFN method (instead of a constant) was explored by Auroux and Blum (2008) and Ruggiero et al. (2015).

> *Ruggiero G. A., Ourmières Y. , Cosme E. , Blum J., Auroux D., and Verron J., 2015: Data assimilation experiments using diffusive back-and-forth nudging for the NEMO ocean model. Nonlin. Processes Geophys., 22, 233–248, www.nonlin-processes-geophys.net/22/233/2015/ doi:10.5194/npg-22-233-2015*

**RC1**: *The use of a constant here bears more resemblance to the methods used in the mathematical field focused on the "synchronization of chaos" (for example see works from Pecora and Carroll, or work by Abarbanel et al.). In those works, however, typically the observations (while they may be sparse in space) are available frequently in time. This scenario seems more akin to an observing buoy in an ocean DA context (unlike, for example a satellite measurement or Argo surfacing profiler).*

> *Louis M. Pecora, Thomas L. Carroll; Synchronization of chaotic systems. Chaos 1 September 2015; 25 (9): 097611. https://doi.org/10.1063/1.4917383*

> *Henry D. I. Abarbanel, Nikolai F. Rulkov, and Mikhail M. Sushchik*
> *Phys. Rev. E 53, 4528 – Published 1 May 1996*

*These works largely used nudging methods to achieve synchronization and are highly related to DA methods. For example, Penny (2017) showed the connections between modern DA methods and concepts from synchronization of Chaos:*

> *Penny, S.G.; Mathematical foundations of hybrid data assimilation from a synchronization perspective. Chaos 1 December 2017; 27 (12): 126801. https://doi.org/10.1063/1.5001819*

**AC**: We thank the reviewer for pointing out the parallel between nudging and synchronization.

LINE 110,

**RC1**: *I think it would be helpful to plot what the eta coefficient looks like as defined by equation 6.*

**AC**: We have included a plot of eta in the revised manuscript.

**MC:** See Line 127, Equation 6.

**RC1**: *It is a bit unclear from the equations (5) and (6) - is the departure X_obs-H(X) the input argument to the function eta(x) as defined in equation (6), or is eta a constant that is the nudging coefficient applied to the departure? If the latter, what is the input 'x' value to eta(x)?*

**AC**: The former: the departure is the input to the function eta(x) in Equation 6.

**MC:** See Lines 124-127. Revised sentence to clear any confusion.

LINE 115, *"if the results can still be achieved with sparse observations"*

**RC1**: *I'd be interested if this investigation includes both sparsity in space and in time.*

**AC**: This investigation did include several experiments that were tested with sparsity of observations in both space and time. See Table 4 (pg. 14) and Table 6 (pg. 17). The results from experiments with even fewer observations are not shown in this manuscript as they did not provide any further valuable information than the experiments included.

**MC:** See Line 70. This sentence has been moved to the Introduction. Table 4 is now on pg. 17 and Table 6 is now on pg. 18.

**LINE 148-161**,

**RC1**: *It might be worth justifying the interpretation of the time units (approx. time) for each model, e.g. the Lorenz-96 timescale is based on the rate of error growth in operational wether prediction models of the time, and was described by Lorenz (1996).*

**AC**: The unit of time used for Lorenz 96 and Lorenz 05 within this manuscript follows from Lorenz (1996, 2005); Lorenz and Emanuel (1998) which states that the scaling in the equations makes the time unit equal to 5 days. The manuscript has been revised to include these references. Although the Lorenz 63 model may have a slightly different time scale (see citation below), Lorenz (1963) states the model uses a dimensionless time unit, and we have made a simple assumption that a time unit in all three Lorenz models used in this study corresponds to 5 days.

> *Ngodock, H. E, S. R. Smith, and G. A. Jacobs, 2009: Cycling the representer method with nonlinear models. In Data Assimilation for Atmospheric, Oceanic and Hydrologic Applications, S. K. Park and L. Xu, (eds.), pp. 321–340, Springer*

**MC:** See Lines 172-178. Revised paragraph and provided references.

**LINE 169, FIGURE 1**, On the panel (b) it says: ***"Truth is shown in teal whereas the orange line is a test run with no DA that started with the same background initial condition."***

**RC1**: *There should have been some difference to cause the divergence in the nature run 'truth' and the free run. In the text (L166-168) it says figure 1b is the first two months of truth - so it does not look like they have the same initial condition. Some more clarity is needed here to describe the experiment setup.*

**AC**: Correct. They do not have the same initial condition. This paragraph and caption have been revised for a clearer explanation. See next comment.

**MC:** See Lines 180-188, "4.1 Lorenz 63 model initialization", as well as the caption for Figure 2.

**LINE 170**,

**RC1**: *Again, for Lorenz 96, the exeriment setup is a bit unclear. You are running the nature run for 1 year and then using the end of the first year as the initial state for the DA experiment. Does that mean the initial 'true' state for the DA experiments, or the background first guess to be corrected to a different truth that is sampled by observations?*

**AC**: The initial state for the DA experiments would be the latter, to be corrected through assimilating sampled observations.

We have revised the text and included a diagram of the experiment setup. The model state after 1-year spinup is used as the initial condition for the DA model experiments. The model spinup continues for an additional 8 months to provide an initial condition for the true model. This continuation of the spinup is to ensure that the two initial conditions are significantly different. The initial condition for the true model is integrated forward without any assimilation for a period of four months, this is referred to as the truth, and is used for sampling observations and evaluating the accuracy of the experiments for each DA method.

**MC:** See Lines 190-198, "4.2 Lorenz 96 model initialization", as well as the model setup diagram in Figure 1. We have also revised "4.3 Lorenz 2005 model initialization", see Lines 200-209.

**LINE 179, FIGURE 2 CAPTION**, Typo:

**RC1**: *'Truth IC' and 'DA IC' - the first apostrophe is backwards in both cases.*

**AC**: We appreciate this comment and have corrected all apostrophes within the manuscript.

**MC:** See Figures 2-4.

**LINE 190-192**, *"Here, we provide a few remarks. The first is that the "best choice" for the value chosen can be different depending on the model being used. There are other cases discussed in the results section below where the optimal value had to be changed to adapt to the parameters given."*

**RC1**: *I think it should be mentioned here or earlier in the text that the "best choice" coefficient is typically derived in modern DA methods as a matrix formed by a combination of information about the background error and observation error. The nudging approach here assumes a diagonal error covariance in both and simply replaces the ratio of background to observation (or more accurately summed) error with a simple constant coefficient.*
    *It would be interesting to take a step closer to a more realistic application by having two or more sets of data with different observation errors associated with them, or to account for uncertainties in the model by expanding the constant coefficient to a full Kalman gain matrix.*
    *In the latter case, the nudging techniques may be more effective in the presence of sparse data if the time-dependent Kalman gain is provided and reasonably accurate. The authors might consider reviewing the Ensemble Transform Kalman-Bucy filters proposed, for example, by:*
        *Amezcua, J., Ide, K., Kalnay, E. and Reich, S. (2014), Ensemble transform Kalman–Bucy filters. Q.J.R.*
        *Meteorol. Soc., 140: 995-1004. https://doi.org/10.1002/qj.2186*
*The relevance of this method is probably closer to that of the compared CCN method.*

**AC**: This section has been updated to remove the terminology 'best choice' to clarify the intent here. We are only discussing the nudging coefficient that gave the best results from the evaluated options presented within this manuscript. We are not trying to estimate the optimal nudging coefficient for the method. That has been done in prior studies with the nudging method in general (Zou et al. 1992) and Auroux et al. (), Ruggiero et al (). We do not think that a full Kalman gain is needed as the nudging coefficients in this study, since that would turn the nudging into a 3DVAR-type DA.

**MC:** This paragraph has been removed. Also see Lines 211-212.

**LINE 201**, *"Several experiments are carried out with different lengths of DA* [analysis] *windows. The length of the forecast is the same as the time window chosen for* [the] *DA* [analysis]*."*

**RC1**: *Please be precise when discussing the DA "analysis window" or DA "analysis cycle window". The term "DA window" is unclear, as it could mean the entire DA experiment period since DA is typically a cycled process.*

**AC**: The intent was the entire DA experiment period and terminology has been updated to reflect this.

**MC:** See Lines 229-230.

**LINE 218, FIGURE 5**,

**RC1**: *It seems that the results here are combining the DA spinup period, the DA performance period, and the forecast into a single assessment. Since spinup for DA methods is a problem in its own right, I'd suggest the authors compare these methods separately - (1) how well the model spins up the state estimate to be close to the true state, and separately (2) how well the algorithms perform once the systems are spun up, and (3) the skill of the resulting forecasts.*

*For example, it has been shown the EnKF methods often take longer to spin up since the ensemble members themselves have to stabilize and converge to the unstable manifold of the system, but after that point can have similar accuracy to more sophisticated variational methods like 4D-Var.*

*Since in practice the spinup is usually only performed once, this does not seem to have high relevance to an operational environment. Rather, it is more interesting to know how the systems perform after this spinup is achieved.*

*That being said, the caption indicates: "All DA experiments assimilated all observations (i.e., all grid points at every timestep/6 minutes)." In that case, I would be very interested to know more about the spinup process, and how robust it is to sparsity in the observations in both space and time.*

**AC**: The spinup process is only performed once before starting the experiments. The spinup itself is not included in any of the results and does not include any assimilation of data. While the period implementing DA and the forecast period are on the same figures and tables, they are calculated independently, and results shown are separated between the two periods (DA and Forecast).

**MC:** See Lines 162-172. We have revised the section explaining the model initializations and spinup period.

**LINE 221**,

**RC1**: *Backwards apostrophe on 'lgp2ts'. Also in Table 2 caption 'all obs' and others.*
**RC1**: *The results in Table 2 are difficult to interpret since thy appear to combine MAE of the spinup and performance periods.*

**AC**: We appreciate this comment and have corrected all apostrophes within the manuscript.
None of the DA performance assessments contain the spinup.

**MC:** See previous comment about spinup period. We have also added included text describing how the results are presented (See Lines 233-244).

**LINE 233**,

**RC1**: *"CCN did not do well with even* [fewer] *observations"*

**AC**: This paragraph has been removed from the manuscript. See next comment.

**LINE 237-238**, *"The conclusion from these results was that a larger nudging coefficient was needed for D-BFN in cases with sparse observations and/or longer time windows."*

**RC1**: *I wouldn't consider skipping observations for 1 timestep (as in Table 3), within the range of linear dynamics of the model, to be 'sparse observations'. Not until the results of Table 4 would this characterization be more appropriate.*

**AC**: We agree with the reviewer that skipping one timestep does not consist of sparse observations. This conclusion was for an experiment whose results were not included within this manuscript. After consideration, we have decided to remove this paragraph from the manuscript.

**MC:** Removed paragraph. See Line 268.

**RC1**: *A comparison to a simple 3D-Var method (using a reasonable background error covariance B matrix) could be useful as a benchmark. If the target is to proceed a method that can be somewhat competitive with 4D-Var, then it seems fair to at least reference a simple 3D-Var as a benchmark, if not 4D-Var itself.*

**AC**: A comparison to a conventional variational method is not included in this manuscript because the comparison has already been evaluated in the reference below. Results are shown for three cases of comparisons with 4DVAR.

> *Auroux, D. and Blum, J.: A nudging-based data assimilation method: The back and forth nudging (BFN) algorithm, Nonlin. Proc. Geophys., 15, 305–319, https://doi.org/10.5194/npg-15-305-2008, 2008.*

LINE 300-303, *"While D-BFN is able to retain accuracy for observations that are sparse in time, due to the advantage of spreading these corrections through the back and forth iterations, we observed that the results from CCN decayed as the density and/or frequency of observations were reduced. "*

**RC1**: *This is a reasonable conclusion, but I'd like to see it demonstrated a bit more rigorously. For example, a full grid search of different combinations of sparsity of observations in both space and time, or with the addition of increased observational noise, and how the methods respond in the 'ideal' scenarios and the more extreme sparse and noisy observation scenarios.*

**AC**: We did explore other sparse distributions of observations (results not shown) beyond what was presented in the manuscript. Our general conclusion is that the accuracy of the methods degrades with the sparsity of the observations, with the Larios and Pei's method degrading faster than the BFN. We do not think that further experiments with sparse observations will alter the results and conclusions of the manuscript. Also, the Larios and Pei method did not do well with the sparse perfect observations. Adding noise to the observations would not help the method gain accuracy.

**Reviewer Comments 2 (RC2)**

**General Comments**

I have basically the same general comments as the other reviewer. The introduction would benefit from more depth and context. More importantly, I think the inclusion of 1-2 other data assimilation (DA) methods, especially at least one filter, is essential. The introduction begins by highlighting the distinction between filters and smoothers, then analyzes two smoothers showing that the one which is more costly (in terms of both person and computational effort) outperforms the other. This result is perhaps unsurprising, but there is very little discussion about when and why this cost-performance tradeoff might be acceptable beyond showing that longer assimilation windows reduce the tradeoff. Including other approaches, especially filters, as counter-examples could provide additional context. Perhaps the less costly but less performant smoother still outperforms a well-calibrated filter. Perhaps it doesn't. The results of such a comparison would go a long way to helping the reader understand when and why they may want to chose one DA method over the other. Finally, given the methodological development in the paper, it is unclear if these approaches can be applied in cases where the observations are not full-rank. Since this is very often the case, some discussion needs to be provided about how one would confront observations that are a lower-rank subset of the state space.

> **Author's Comment (AC):** The objective of this study is to compare two DA methods that were chosen because of ease of implementation. Prior studies have compared DA methods (Ngodock et al. 2006). It is true that we could not find a comparison of the CCN method with classic DA such as 3DVAR, 4DVAR or EnKF, but the BFN has been compared to 4DVAR (cited reference). The two methods chosen for this study have not been compared to each other, and are not representative of all classes of DA. Also, we have revised the introduction to focus on sequential and non-sequential methods, rather than filters and smoothers.

> **Manuscript Change (MC)**: Line numbers are referenced from updated manuscript.

**Specific/Technical Comments**

LINE 14, **"generally two classes"**: *There are other ways to cut the DA pie. For example, variational vs. ensemble.*

> **AC:** We have revised the text to focus instead on sequential and non-sequential methods, as well as reflect that there are other options to categorize DA methods.

> **MC:** See Lines 15-18, as well as Introduction.

LINE 15, **"model background state"**: *Very technical point here, but the model background state is typically conditioned on all past observations and the description up to this point seems somewhat misleading regarding this point.*

> **AC:** We agree with the reviewer that "the model background state is typically conditioned on all past observations". However, the statement in the manuscript is still true, that the DA computes an analysis given a background and observations, regardless of the fact that the background is conditioned on past observations.

> **MC:** See Lines 19-20. Removed "state".

LINE 21, **"suppressing the time variability in the observations"**: *This isn't really true. The 3D-Var systems used at many DA centers use interpolation across the beginning, middle, and end of the time window. This makes the comparison to filters here seem like a bit of a straw man. To address this concern, I would very much like to see an example of 3D-Var with this type of time interpolation compared to the methods presented in this paper.*

> **AC:** We are aware of the FGAT and the implementation that the reviewer mentions. However, 3DVAR cannot assimilate observations taken at the same location but different times within the observations or assimilation windows, because the method can only produce an analysis of the state at one time. As mentioned earlier, a

comparison of the BFN with 4DVAR has already been published, so there is no need to compare the methods here with 3DVAR.

**MC:** See Lines 25-29. Classification changed from filter to sequential. Added the distinction of intermittent and continuous addressing the suppression of the time-variability.

**LINE 39**, **"seems to"**: *I would hope we could be more precise in peer-reviewed scientific literature.*

**AC:** The text has been revised.

**MC:** See Line 47. Changed "seems to converge" to "converges".

**LINE 44**, **"BFN can however"**: *I would suggest perhaps a new paragraph here and a few more details about AOT. There are a lof of acronyms introduced in a short time (BFN, CDA, AOT, CCN, etc.), and it would best to either reduce or make the distinctions between all of these clear. See also comment on lines 117-119.*

**AC:** We have revised the text to start with a new paragraph and have removed any acronyms not used outside of the introduction. We have also included brief details about the AOT method.

**MC:** See Lines 49-60. Started a new paragraph, edited, and moved the referenced sentence to the end of the paragraph (Lines 59-60) and added details about AOT (Lines 51-54). We also removed several acronyms throughout the Introduction that were not essential in the remaining text: KS, CDA, KSE, TLM. Included the method name "Concave-Convex Nonlinearity (CCN)" within the introduction.

**LINES 78-79**, **"M is used ... the forcing"**: *I don't understand why this is worth noting. M and F are just variable names.*

**AC:** We agree it is not necessary and have updated the manuscript.

**MC:** See Line 92.

**EQUATION 2**: *How would this be applied if H is not full rank? In fact, it seems like H is assumed to be the identity matrix here. The latter would only be true in test cases and never in practice.*

**AC:** We disagree with the reviewer. Full rank or not, H is the observation operator and is not assumed to be the identity matrix.

**LINE 86**, **"state"**: *Again, this is the observation operator applied to the state and is only the state when H is the identity.*

**AC:** The text has been revised. The comparison of the observations and the model is only done at the observation locations.

**MC:** See Lines 99-100. Added "at the observation locations".

**LINE 95**, **"2.1.1"**: *Suggest 2.2*

**AC:** We appreciate this find and it has been corrected.

**MC:** See Line 108.

**LINE 96**, **"linear AOT method"**: *I don't really understand what this is and some description would help.*

**AC:** A brief description of the AOT method has been added to the introduction and more details have been included in this section.

**MC:** See Lines 51-54 (Introduction) and Lines 109-120.

LINES **97-98**, ***"The first approach ... linear AOT method"***: *I don't think there's been enough introduction to this point for the reader to have a sense of why this is meaningful.*

**AC:** See reply to previous comment.

**MC:** See Lines 51-54 (Introduction) and Lines 109-120.

LINES **101-102**, ***"Concave-Convex Nonlinearity"***: *Acronym has already been defined. Suggest either writing CCN or including the abbreviation again.*

**AC:** We have adopted the suggestion to abbreviate again.

**MC:** See Line 117.

LINE **110**, ***"eta(x) = eta_3(x)"***: *This additional notation seems to not be needed.*

**AC:** The additional notation is used in the referenced paper where they present three methods. We agree that it is not relevant within this manuscript and the name of the method alone will allow readers to distinguish the method we are using.

**MC:** See Line 127. Also removed the reference in the text to eta_3.

LINES **116-117**, ***"not explicitly stated ... every timestep"***: *Seems like it would be better to contact the authors rather than speculating.*

**AC:** We have removed this sentence from the manuscript.

**MC:** See Line 132.

LINES **117-119**, ***"This paper investigates ... as in BFN."***: *This seems introductory and maybe should go in the Introduction.*

**AC:** We agree with the reviewer and have adopted the suggestion to move it to the introduction.

**MC:** See Lines 68-71 in the Introduction.

LINES **136-137**, ***"For clarification ... D-BFN"***: *Again, these are just variable names. This note seems to conflate the variable name with the thing it denotes.*

**AC:** We agree that this clarification is not necessary and have removed the sentence from the manuscript.

**MC:** See Line 149.

LINE **155**, ***"create sufficient chaos"***: *I have no idea what "sufficient chaos" is.*

**AC:** This sentence has been revised to reflect the statement in Lorenz (2005) and Lorenz and Emanuel (1998) that the models require a period of spinup to remove any transient effects.

**MC:** See Lines 162-163.

LINE 162, **"6 minutes"**: *I'm confused what these times correspond to. Is this walltime or the time variable of the equation.*

**AC:** This is not the wall time; this is the equivalent unit of time for the timestep. The table has been revised to remove the approximate time from its own column and has instead been added within the timestep size column (i.e., $dt = 1/40 = 3$ hours).

**MC:** See Table 1. Revised table to remove confusion.

LINE 165, **"significantly different"**: *Suggest either being more precise or removing this comment.*

**AC:** We agree with this suggestion and have revised the text to be more precise.

**MC:** See Lines 185-187.

FIGURE 1: *While it's spelled out in the caption, an inset legend in the figure might be helpful.*

**AC:** We agree and have updated the figure in the manuscript to include a legend.

**MC:** See Legend in Figure 2.

FIGURE 2, **"the top figure"**: *Suggest using the term "panel" here and elsewhere instead of "figure" since the entire thing is "Figure 2".*

**AC:** Figure captions have been revised in the manuscript.

**MC:** See all figure captions. Revised to use (a), (b), ... instead of 'top figure'.

FIGURE 4: *The figure caption and labeling are insufficient for me to understand what's going on. Does each segment in panels a and c correspond to the entire frame of panels b and d? Please clarify.*

**AC:** Figure captions have been revised in the manuscript.

**MC:** See all figure captions. They have been revised to be clearer.

LINE 221, **"1gp2ts"**: *Seems like these acronyms could be improved. Perhaps 1GP-2TS?*

**AC:** We have adopted this suggestion and revised the manuscript.

**MC:** See all Tables and Figures in the results section (5).

LINE 231, **"Other experiments ... should be discussed"**: *If the experiments were relevant they should be provided, perhaps in a supplement/appendix.*

**AC:** We agree that it does not provide relevant information to the comparison of the two methods and have removed it from the manuscript.

**MC:** See Line 268.

TABLE 2, **"Fcast"**: *Suggest "FC" for consistency with "DA".*

**AC:** We thank you for the suggestion and have adopted this change.

**MC:** See all Tables and Figures in the results section (5).

TABLE 2, *all obs, CCN, 1m and 2m Fcast: I don't understand how forecast errors can be smaller, and so much so, than the DA errors. This seems like a typo/mistake. Could you please explain? See also Table 6.*

**AC:** We understand the reviewer's confusion as the text lacks clarity. The forecast discussed here is not the one used as the background for assimilation. This forecast is computed after the entire assimilation experiment is completed, and benefits from an accurate initial state at the end of the assimilation experiment. On the other hand, the DA experiment started from a very wrong initial state and even though the assimilation did improve the model accuracy by the end of the assimilation experiment, the analysis errors averaged over the entire assimilation experiment are still impacted by the high initial errors at the beginning of the assimilation experiment. We note that this only happened in the particular case of 1-month assimilation of all observations using the CCN method.